# Self-Efficacy Mediates Acculturation and Respite Care Knowledge of Immigrant Caregivers

**DOI:** 10.3390/ijerph182010595

**Published:** 2021-10-10

**Authors:** Shu-Fen Kuo, I-Hui Chen, Tsai-Wei Huang, Nae-Fang Miao, Kath Peters, Min-Huey Chung

**Affiliations:** 1School of Nursing, College of Nursing, Taipei Medical University, Taipei 11031, Taiwan; sfkuo6@tmu.edu.tw (S.-F.K.); ichen4@tmu.edu.tw (I.-H.C.); tsaiwei@tmu.edu.tw (T.-W.H.); 2Post-Baccalaureate Program in Nursing, College of Nursing, Taipei Medical University, Taipei 11031, Taiwan; naefang@tmu.edu.tw; 3School of Nursing and Midwifery, Western Sydney University, Penrith, NSW 2751, Australia; K.Peters@westernsydney.edu.au; 4Department of Nursing, Shuang Ho Hospital, Taipei Medical University, New Taipei City 23561, Taiwan

**Keywords:** respite care knowledge, immigrant caregivers, self-efficacy, acculturation

## Abstract

Past studies have shown that acculturation and self-efficacy can affect respite care knowledge, which are notable issues among immigrant caregivers due to the rapid increasing aging family members. The aim of this study was to investigate relationships among acculturation, self-efficacy, and respite care knowledge in immigrant caregivers, and to determine the mediating effects of self-efficacy on the relationship between acculturation and respite care knowledge. A cross-sectional design was used. We enrolled 134 female immigrant caregivers who had married Taiwanese men and lived with care recipients who used LTC services. Based on Baron and Kenny’ mediating analytic framework, multiple regression and Sobel tests were used to examine whether self-efficacy mediated the relationship between acculturation and respite care knowledge. The findings showed that after controlling for confounding factors, acculturation and self-efficacy separately affected respite care knowledge (B = 0.229, standard error (SE) = 0.084; B = 0.123, SE = 0.049, respectively). Acculturation had a positive impact on respite care knowledge through self-efficacy (B = 0.181, SE = 0.084). Therefore, self-efficacy partially mediated the effect of acculturation on respite care knowledge, and accounted for 20.9% of the total mediating effect in this study. Acculturation predicted immigrant caregiver’ respite care knowledge partially through self-efficacy. The association between acculturation and respite care knowledge was partially mediated by immigrant caregivers’ self-efficacy. As a result, it was proposed that boosting self-efficacy could increase and drive immigrant caregivers’ respite care knowledge. To assist this population in obtaining enough resources, targeted educational programs to promote immigrant caregivers’ self-efficacy should be designed and implemented. Furthermore, health care practitioners should be aware of the relevance of immigrant caregivers’ acculturation.

## 1. Introduction

People aged 65 years or over accounted for more than one-fifth of the population in 17 countries of the world (7.2%) in 2019 [1]. Elderly people normally face more disease and disability problems. Therefore, the increasing needs of caregivers, which include spouses, daughters, and daughters-in-laws, are the first priority in aging and disability care [2]. The results from a meta-analysis reported the prevalence of depression and being burdened in caregivers as 31.24% and 49.26%, respectively [2]. The aim of respite care services is to reduce burdening incidents and to help prevent the burden on family caregivers [3,4]. Caregiver participants in previous research agreed that accessing respite care services was beneficial to both care recipients and themselves [5], and reported a positive sleep quality when care recipients are admitted for a temporary residential respite period [6]. If caregivers know what the respite care is and understand their variety of forms, they are better able to make effective use of respite care services [7]. However, past studies have shown that 40% of caregivers do not use respite care service resources [8]. Therefore, given the substantial effect of resources on care recipients’ outcomes, understanding the meaningful component of these resources is vital, especially regarding respite care knowledge [9].

Globally, there were 272 million international migrants, accounting for 3.5% of the world’s population, in 2019 [10]. There are increasingly more immigrant caregivers who live with care recipients in the caregivers’ home than white caregivers in the US, and they have exhibited significant declines in self-related health because of less use of respite care than white caregivers during the past 5 years [11]. To date, the immigrant caregivers’ service utilization has focused on issues such as not using home care-based services [9] and the burden of family caregiving [3,12]. Furthermore, immigrant caregivers typically navigate services on behalf of their care recipients, and tend to use the long-term care (LTC) services when they gain adequate knowledge and access to those resources [13]. However, immigrant caregivers do not sufficiently use healthcare services, possibly because of a much lower level of knowledge [13,14,15], which could be caused by adaptation processes in cross-cultural environments [9]. There is a gap in the literature, such as how acculturation impacts the perceived knowledge of respite care services. To address this gap, a quantitative study was used to assess the mediation of self-efficacy between acculturation and respite care knowledge in Taiwan.

## 2. Literature Review

### 2.1. Respite Care Knowledge

An extensive amount of literature has explored the determinants of health behaviors among immigrants, including self-efficacy and acculturation [16,17]. In addition, a past study showed that social and contextual factors were mediators between acculturation and health behaviors [18]. Some studies on health services knowledge found that related factors of increased immigrants’ knowledge were language ability and education level, which impacted the acculturation process [19,20]. The gaps in respite care services access were due to the lack of understanding across multiple service domains [21]. On policies regarding respite care services, the familiarity of multiple service domains was important among immigrants, for example respite at home and institutional respite. However, there were large differences regarding health knowledge between immigrant groups, which need more understanding regarding the relevant factors of immigrants’ knowledge [22]. Nevertheless, few studies have examined the relationship of acculturation and self-efficacy on LTC knowledge outcomes, especially respite care knowledge. Accordingly, the purpose of this study was to explore the relationships of acculturation and self-efficacy with respite care knowledge among Asian immigrant women in Taiwan, and to determine the mediating relationship s of self-efficacy between acculturation and respite care knowledge.

### 2.2. Acculturation and Self-Efficacy

Healthcare professionals and social workers have identified some reasons for the lower utilization of healthcare services by immigrant caregivers in past studies, including language barriers, lack knowledge of healthcare services/systems, and the perception of not needing such services [9,23]. The World Health Organization further proposed a conceptual framework of social determination of health, which pointed out that racial discrimination and exclusion often significantly worsened a person’s health status. So, culture and societal values are reasons for health inequalities [24]. Findings from a previous study showed that acculturation was positively related to immigrants’ service expectations, which were very important for their service utilization [25]. The definition of acculturation is “the process of cultural and psychological change that takes place as a result of contact between cultural groups and their individual members” [26], and immigrant caregivers need to adapt in the new rules and values of the host country through the acculturation process [10]. Two factors are used as proxy measures of acculturation: the length of stay and the main host language proficiency [18,27,28]. For instance, one study showed that immigrants living in the US for more than 10 years seemed to have adapted to Western points of view regarding elder care, and were higher ratios to use a nursing home for better health care [29]. Another study explored how Spanish-speaking immigrants had less oral health knowledge due to their poor English ability as they were unable to understand the promotional material [30]. The above-mentioned studies demonstrate that acculturation could be a determinant of immigrants’ use and knowledge of health services.

Psychosocial and educational interventions are effective at improving self-efficacy and health knowledge among married immigrants, which is important as self-efficacy is a personal resource factor that facilitates coping [31,32]. Furthermore, self-efficacy and service knowledge among immigrant caregivers have been associated with smaller caregiver burdens and better health outcomes for care recipients [9]. Self-efficacy is defined as “a person’s particular set of beliefs, and refers to the amount of confidence held that an individual can execute actions to accomplish desired goals in specific situations“ [33].

### 2.3. The Mediating Role of Self-Efficacy

Previous research found that immigrant women’s health beliefs depend on their social and cultural experiences upon arrival in the host country, which determine their sense of self-efficacy of getting health service information and gaining access to health services, such as mammography screening [34]. Further research also identified that higher levels of self-efficacy and language proficiency of original country were related to lower levels of food insecurity among immigrants [35], and that overall acculturation levels in the host society predicted self-efficacy in international graduate students [36]. Additionally, there has been some discussion of the potential mediating mechanisms of social and contextual factors in the relationships between acculturation and health outcomes in past immigrant studies [18]. Accordingly, self-efficacy has been viewed as a contextual factor and should be further explored for its potential mediating role. A mediation model is proposed (Figure 1). This is the first study to exclusively address the mediating role of self-efficacy between acculturation and respite care knowledge among Asian immigrant women.

## 3. Methods

### 3.1. Study Design and Participants

The cross-sectional study was designed among immigrant women in Taiwan in September to December 2019. This study used a convenience sampling method to recruit participants. First, six areas were selected in Taiwan, namely Taipei, Taoyuan City, Changhua County, Tainan, Kaohsiung, and Hualien County. Second, we approached supplementary schools and non-governmental organizations in these six areas. Then, we recruited school teachers and social workers from participating institutions, and held workshops for them to comprehend the study purpose, procedures, questionnaires, and research ethics. Finally, trained school teachers and social workers explained the study aims to and received informed consent from immigrant women, who will complete a questionnaire in their native language in 15~30 min. This project was reviewed and approved by the author university’s Joint Institutional Review Board (no. N201903144). All participants could voluntarily participate in this study and withdraw from this study at any stage.

The participants’ inclusion criteria were as follows: (1) capable of communicating in Mandarin or Taiwanese, (2) married to a Taiwanese man, (3) able to read the native language of their nationality, and (4) living with care recipients who used LTC services. The exclusion criteria of participants were as follows: (1) undergoing divorce proceedings, (2) being resettled after experiencing domestic violence, and (3) refusing to accept an interview or respond to the questionnaires.

The criteria of sample size calculation with a linear multiple regression for statistical significance were included a power of 0.80, an effect size of 0.15, alpha of 0.05, and number of predictors of 11. The total sample size needed, at least, was 123. The study initially approached 144 immigrant women. Ultimately, 134 married immigrant women completed and returned the questionnaire. The response rate was 93.0%.

### 3.2. Data Collection

Demographic characteristics were treated as confounding factors in this study. Demographic factors of immigrant caregivers included nationality, age, education received in the home country, education received in Taiwan, citizenship identity, family structure, monthly income, living environment, and sources of information on LTC services. The living environment was reported as urban (including industrial areas) or rural countryside (including fishing villages). Sources of information on LTC services were measured using the number of LTC information resources, which included TV, the internet, hospital discharge preparation office, LTC manager center, 1966 special phone line, resident office/district office, relatives/friends, and institutions (associations and foundations). One source got one score, and a higher score indicated more message sources of information on LTC services. The other instruments are shown below.

#### 3.2.1. Respite Care Knowledge

Respite care knowledge was treated as a continuous variable. There were five items asked of immigrant caregivers regarding how familiar they were with respite care services, including respite at home, institutional respite, daycare center respite care, small multifunctional institutional respite, and special alley station respite. Each item was measured using a four-point Likert scale (1 “not familiar” to 4 “very familiar”). A higher score indicated a higher level of respite care knowledge. It was 0.94 of Cronbach’s α in this study.

#### 3.2.2. Acculturation

Acculturation was tested by the language spoken at home and the years in the host country [27,28]. A score was given for language ability in the host home: speaking and writing Chinese well (2 points), speaking Chinese only mildly (1 point), or speaking Taiwanese only poorly (0 points). A score was given for living in the host country for >10 years (2 points), 6~10 years (1 point), or 0~5 years (0 point). Finally, a combined score of 0 indicated the lowest level of acculturation, while a score of 4 indicated the highest level.

#### 3.2.3. Self-Efficacy

The Chinese version of the General Self-Efficacy Scale (CSE) was developed on the basis of Bandura’s self-efficacy theory and to assess self-efficacy in response to stressful events by self-reported answers. There are 10 items on the CSE scale with a four-point Likert scale (1 “not at all true” to 4 “exactly true”). The scale has been widely used in many countries, including China, Japan, South Korea, and India. The total score ranges 10~40, with a higher score indicating a greater self-efficacy. The Cronbach’s α of the CSE in this study was 0.91, which is considered to be a high reliability compared with the 0.89 from a past study [37].

### 3.3. Statistical Analyses

Descriptive analyses were expressed as number (percentages) and mean (standard deviation, SD). An analysis of variance (ANOVA), *t*-test, and Pearson’s correlation were utilized to examine the differences in the relationships of demographics, acculturation, and self-efficacy with respite care knowledge. Self-efficacy was examined as a mediator of the correlations between acculturation and respite care knowledge, and using a multiple regression analysis. The mediating analytic framework in Figure 1 was described by the Baron [38] and Kenny’ analysis plan [38]. The significant confounding factor was first put into the regression model to control the effect on respite care knowledge, which was age and sources of information on LTC services. The mediating analytic was performed to investigate if (1) acculturation significantly predicted respite care knowledge (Path C in Figure 1), (2) acculturation significantly predicted self-efficacy (Path A in Figure 1), (3) self-efficacy significantly predicted respite care knowledge (Path B in Figure 1), and (4) acculturation significantly predicted respite care knowledge after controlling for self-efficacy (Path C’ in Figure 1). The mediating effect of self-efficacy was partial or complete mediation, depending on the regression correlation coefficient of Path C’ with or without statistical significance. The mediation effect value was calculated as axb, and the ratio of the mediating effect with the total effect was axb/c (a, b, c, and c’ are the regression correlation coefficient of path A, path B, path C, and path C’, respectively) [39].

The effect of the mediating variable was assessed by the Sobel test, which was used to test the indirect effect of the acculturation on respite knowledge via self-efficacy [38]. As in Figure 1, the path from acculturation to self-efficacy is denoted as a and its standard error is $s_{a}$; the path from self-efficacy to the dependent variable is denoted as b and its standard error is $s_{b}$. The exact formula of the Sobel test is shown below [38]:(1)$$z\;=\frac{a\times b}{\sqrt{b^{2}s_{a}{}^{2}+a^{2}s_{b}{}^{2}}}$$

Finally, statistical significance was defined as two-tailed *p* < 0.05. All data analyses were performed using SPSS 20.0 (SPSS Inc., Chicago, IL, USA) software.

## 4. Findings

### 4.1. Characteristics of Participants

The final sample was composed of 134 immigrant women, among whom 50.7% were from Vietnam; 20.9% were from China; 16.4% were from Indonesia; and 12% were from Thailand, the Philippines, and Cambodia. The average age was 39.05 (SD = 8.99) years and average length of staying in Taiwan was 12.6 (SD = 8.68) years. Regarding education received in their home country, 58.2% had attended senior high school or above. As for education received in Taiwan, 9.0% had attended senior high school or above. There were 3.7% who only spoke Taiwanese, 56% who only spoke Chinese, and 40.3% who spoke and wrote Chinese, indicating that less than half of the immigrants could not deal with documents in the national Chinese language. In addition, more than half of the immigrant women were members of a nuclear family; 65.7% lived in a family with a monthly income of <NT$30,000 (≈US$1000) and 64.9% lived in urban areas. The mean scores on the average of information sources of LTC services, acculturation, and self-efficacy were 1.84 (SD = 1.13), 2.70 (SD = 1.10), and 2.51 (SD = 0.66), respectively. The average score of the respite care knowledge was 2.61 (SD = 1.01) (Table 1).

### 4.2. Factors Associated with the Respite Care Knowledge

Table 2 shows that age (*r* = 0.292, *p* = 0.001), sources of information on LTC services (*r* = 0.220, *p* = 0.008), acculturation (*r* = 0.351, *p* < 0.001), and self-efficacy (*r* = 0.286, *p* = 0.001) were significantly positively related to respite care knowledge. Other factors were not significantly related to respite care knowledge.

### 4.3. Mediation Analysis

Table 3 shows a summary of the mediating role of self-efficacy in the relationship between acculturation and respite care knowledge. The results of the total effect (path C) indicated that acculturation predicted respite care knowledge after controlling for age and sources of information on LTC services (B = 0.229, standard error (SE) = 0.084, *p* = 0.007, 95%CI, 0.063–0.394). Acculturation had a positive impact on self-efficacy (path A) (B = 0.123, SE = 0.049, *p* = 0.018, 95%CI, 0.022–0.224). As shown in Figure 1, both acculturation and self-efficacy had a positive impact on respite care knowledge (path C’, B = 0.181, SE = 0.084, *p* = 0.033, 95%CI, 0.015–0.346; path B, B = 0.391, SE = 0.121, *p* = 0.002, 95%CI, 0.152–0.630). The mediation effect value was calculated at 0.123x0.433, that is, 0.048, and the ratio of the mediating effect over the total effect was 20.9% (0.048/0.229 = 0.209).

In addition, the Sobel test showed that the drop in beta weight in the relationship between acculturation and respite care knowledge was statistically significant (z = 1.982, *p* = 0.047), indicating that self-efficacy had a significant mediating effect on the relationship between acculturation and respite care knowledge. The regression correlation coefficients of path A, path B, path C, and path C’ were all significant. Therefore, self-efficacy partially mediated the effect of acculturation on respite care knowledge. Acculturation predicted immigrant caregivers’ respite care knowledge partially through self-efficacy.

## 5. Discussion

Our findings show that acculturation and self-efficacy positively predicted respite care knowledge among immigrant caregivers. Furthermore, the effect of acculturation on respite care knowledge was partially mediated by self-efficacy when controlling for caregiver age and sources of information on LTC services.

International evidence has shown that immigrant caregivers who use respite care have significantly lower stress and burdens than before using respite care [3]. In this study, the immigrant caregivers’ respite care knowledge was 2.61 (SD = 1.01), which showed that immigrant caregivers’ care recipient knowledge was not enough, even if it was related to their experiences on daily life care events combined with the government’ LTC services. This issue could be possibility extended to more immigrant caregivers if they have not yet received LTC services for care recipients, because the respite care services utilization rate was 12.6% in a Taiwanese LTC service utilization survey [40]. Lower respite care utilization could be related to caregivers’ lower respite care knowledge, especially within the population of immigrant caregivers living with older home-care recipients [9,41]. Sometimes, a married immigrant is a family’s main caregiver, but she may have few opportunities to learn about respite care services in Taiwan because of a poor acculturation process at home [15]. In addition, many immigrant caregivers rely on personal networks to obtain respite care information; however, immigrant women mentioned that although there is no language barrier between friends from the same country, the correctness of the respite care knowledge of those friends also affects immigrants’ access to respite care services [42]. In this study, we found that sources of information on LTC services was positively related to respite care knowledge. Therefore, respite care information could be actively provided to immigrant women by hospital and community nurses during the first year of the immigration period. For instance, both ward nurses caring for older adults and discharge-planning nurses could discuss respite care concepts with immigrant caregivers, and provide brochures and publications regarding LTC services for immigrants.

We also found that acculturation was positively correlated with respite care knowledge. In particular, the language component of acculturation may have a notable effect on respite care knowledge; lower language proficiency could become a barrier as a core challenge that makes it difficult for immigrants to navigate the system and acquire information in the new country [8]. The availability of trained interpreters from migrant communities could be one of key aspects for providing migrant-sensitive care [10]. There are part-time immigrant interpreters at hospitals and community health centers in Taiwan during work days who can satisfactorily help immigrants with poor Mandarin or Taiwanese competency, who are seeking medical attention. However, to date, part-time interpreters cannot sufficiently share information regarding respite care to immigrant caregivers due to a lack of trained interpreters to address respite care in LTC. To ensure that all information is communicated accurately and efficiently, it is recommended that nurses work with interdisciplinary health professionals and provide training to these part-time immigrant interpreters. Meanwhile, LTC service information media and tools, such as brochures and broadcasts, can be developed using multiple languages for new immigrants. Then, clinical nurses could use these multiple-language resources to improve the care quality provided to immigrant caregivers. The length of stay in the host country, another component of acculturation in this study, was found to be a significant factor in the willingness to use services [29]. Nurses in hospitals and community centers should be sensitive to acculturation processes when providing information regarding LTC services, especially respite care services, to immigrant caregivers.

We further demonstrated the positive effect of self-efficacy on the respite care knowledge of immigrant caregivers. Ward nurses in hospitals must be competent in recognizing and responding to issues of immigrant caregivers, and could play crucial roles in increasing immigrant caregivers’ self-efficacy of respite care knowledge. Hence, more on-the-job education is needed from discharge planning center nurse specialists or LTC case managers for ward nurses, especially nurses working with older adults. This is the first study to demonstrate self-efficacy as a partial mediator between acculturation and respite care knowledge. Immigrant caregivers’ self-efficacy with respect to respite care knowledge is the confidence held that one is able to execute respite care actions, which are related to immigrant caregivers’ self-reflection, based on the transformative learning model [43]. Accordingly, self-reflection could be a part of nursing education, which supports immigrant caregivers’ learning [44] and in turn can promote immigrant caregivers’ self-efficacy under the process of migration adaptation. By doing so, it may benefit the health outcomes of care recipients cared for by immigrant caregivers.

### 5.1. Limitations

There are several limitations of the current research. First, the cross-sectional design per se was a limitation on the ability to assess causal relationships among acculturation, self-efficacy, and respite care knowledge. Second, language use and years spent in the host country might not be closely related to other dimensions of acculturation, such as values and identification, which should be examined in future studies. Third, this study used convenience sampling, which limits its generalizability to the population and other immigrant caregivers from Western countries. Fourth, this study considered how familiar they were with five respite care services in Taiwan to measure respite care knowledge. Sometimes, this knowledge could be expanded as respite caregiver knowledge, which allows respondents to measure different aspects of their understanding on respite care meanings from knowing something to will do something [3]. Therefore, asking different questions regarding respite care is needed to complement the self awareness on respite care, such as are recipients’ development, sense of peace and life fulfillment among caregivers, mental health support for the caregiver, and expansion of future vision [45]. So, the respite care knowledge among immigrant caregivers could be more specifically definition in the future study.

### 5.2. Implications of Research Results

Hospital and community nurses, particularly those who care for elderly populations, can regularly assess immigrant caregivers’ self-efficacy under acculturation adaptation processes. By doing so, nurses would be able to early identify groups of immigrant caregivers with lower self-efficacy and provide interventions such as education in order to enhance their well-being and care quality. Further based on this research, nurses can build immigrant caregivers’ educational programs on respite care at hospitals and community health centers. Meanwhile, nurses can work with a coordinator and interdisciplinary health professionals to ensure that these programs meet immigrant caregivers’ needs depending on their cross-culture experiences [16,46]. In addition, nurses themselves should be familiar with strategies to foster self-efficacy in nursing practice, such as self-reflection. Immigrant caregivers with lower levels of acculturation should, as early as possible, be invited to participate in nursing educational programs. Through participating in culturally sensitive nursing educational programs, immigrant caregivers can obtain adequate respite care knowledge.

## 6. Conclusions

This study explored respite care knowledge among immigrant caregivers, and demonstrated that self-efficacy partially mediated the relationship of acculturation with respite care knowledge of immigrant caregivers. As a result, it was proposed that increasing self-efficacy could help immigrant caregivers learn more about respite care. Targeted educational initiatives to increase immigrant caregivers’ self-efficacy should be devised and implemented to help this population receive adequate resources. Furthermore, health care providers should be aware of the importance of acculturation for immigrant caregivers. Considering immigrant caregivers’ respite care knowledge in this study was not enough, even if related to their experiences on daily life care events combined the government’ LTC services, and targeted educational programs to improve immigrant caregivers’ self-efficacy may increase their confidence in respite care knowledge. Moreover, we found self-efficacy to be a partial mediator; therefore, further studies could include other socio-cultural factors, such as social support [18], as mediators to investigate their influences on the relationship between acculturation and respite care knowledge among immigrant caregivers.

## Figures and Tables

**Figure 1 ijerph-18-10595-f001:**
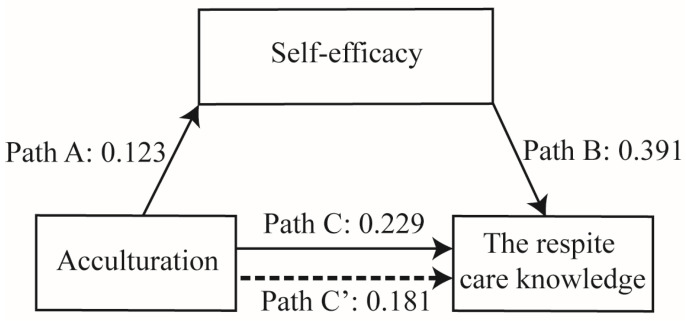
Mediating effect of self-efficacy between acculturation and respite care knowledge. Non-standardized path coefficients are presented. A partial mediating effect was found for self-efficacy.

**Table 1 ijerph-18-10595-t001:** Descriptive analysis (*n* = 134).

Variable	*n* (%)/Mean (SD)
Nationality	
Vietnam	68 (50.7)
China	28 (20.9)
Indonesia	22 (16.4)
Thailand	4 (3.0)
The Philippines	9 (6.7)
Other	3 (2.2)
Age (years)	39.05 (8.9)
Length of stay in Taiwan (Years)	12.6 (8.6)
Education received in home country	
Illiterate or elementary school	20 (14.9)
Junior high school	36 (26.9)
Senior high school	50 (37.3)
University or higher	28 (20.9)
Education received in Taiwan	
Illiterate	69 (51.5)
Elementary school	38 (28.4)
Junior high school	15 (11.2)
Senior high school	6 (4.5)
University or higher	6 (4.5)
Language ability	
Speaking Taiwanese only	5 (3.7)
Speaking Chinese only	75 (56)
Speaking and writing Chinese	54 (40.3)
Citizenship identity	
Resident permit	51 (38.1)
Permanent residence permit	6 (4.5)
Citizen	77 (57.5)
Family structure	
Nuclear family	66 (49.3)
Eclectic family	55 (41.0)
Extended family	13 (9.7)
Monthly income ^a^	
<NT$30,000	88 (65.7)
NT$30,001~50,000	28 (20.9)
≥NT$50,001	18 (13.4)
Living environment	
Urban (including industrial areas)	87 (64.9)
Rural countryside (including fishing villages)	47 (35.1)
Sources of information on LTC services	1.84 (1.13)
Acculturation	2.70 (1.10)
Self-efficacy	2.51 (0.66)
Respite care knowledge	2.61 (1.01)

Note: SD, standard deviation; LTC, long-term care. ^a^ The average exchange rate in July 2021 was AUD$1.00 ≈ New Taiwan (NT)$21.29.

**Table 2 ijerph-18-10595-t002:** Associations between demographics and respite care knowledge (*n* = 134).

Variable	Respite Care Knowledge	
	Mean (SD)	*F* (*p*)/*t* (*p*)/*r*(*p*) ^b^
Nationality		0.927 (0.466)
Vietnam	2.56 (1.05)	
China	2.45 (0.97)	
Indonesia	2.63 (1.00)	
Thailand	2.90 (0.95)	
The Philippines	3.20 (0.81)	
Other	3.00 (1.00)	
Age (years)		0.292 (0.001) **
Education received in home country		2.237 (0.087)
Illiterate or elementary school	2.54 (0.84)	
Junior high school	2.64 (1.12)	
Senior high school	2.40 (1.01)	
University or higher	3.00 (0.86)	
Education received in Taiwan		1.658 (0.164)
Illiterate	2.64 (1.00)	
Elementary school	2.46 (1.00)	
Junior high school	3.09 (1.10)	
Senior high school	2.00 (0.59)	
University or higher	2.66 (0.94)	
Citizenship identity		1.606 (0.205)
Resident permit	2.46 (1.04)	
Permanent residence permit	2.23 (1.02)	
Citizen	2.74 (0.97)	
Family structure		0.061 (0.940)
Nuclear family	2.59 (1.08)	
Eclectic family	2.65 (0.91)	
Extended family	2.56 (1.10)	
Monthly income ^a^		0.150 (0.861)
<NT$30,000	2.58 (1.06)	
NT$30,001~50,000	2.70 (1.02)	
≥NT$50,001	2.61 (0.66)	
Living environment		0.094 (0.925)
Urban (including industrial areas)	2.62 (1.07)	
Rural countryside (including fishing villages)	2.60 (0.89)	
Sources of information on LTC services		0.220 (0.008) **
Acculturation		0.351 (<0.001) ***
Self-efficacy		0.286 (0.001) **

Note: SD, standard deviation; LTC, long-term care. ** *p* < 0.01; *** *p* < 0.001. ^a^ The average exchange rate in July 2021 was AUD$1.00 ≈ New Taiwan (NT)$21.29. ^b^
*F**(p)/t**(p)/**s**(p) =* ANOVA (*p* value)/*t*-test (*p* value)/Pearson’s correlation (*p* value).

**Table 3 ijerph-18-10595-t003:** Summary of the mediating effects of self-efficacy between acculturation and respite care knowledge (*n* = 134).

Effect	Independent Variables	Dependent Variables	B	SE	*t*	*p* Value	95%CI
Total effect (path C)	X	Y	0.229	0.084	2.728	0.007	0.063–0.394
Indirect effect (path A)	X	M	0.123	0.049	2.398	0.018	0.022–0.224
Indirect effect (path B)	M	Y	0.391	0.121	2.033	0.002	0.152–0.630
Direct effect (path C’)	X	Y	0.181	0.084	2.161	0.033	0.015–0.346

Note: the model was in controlling for age and sources of information on LTC services; B, unstandardized coefficient; SE, standard error; X, acculturation; M, self-efficacy; Y, respite care knowledge. Sobel test, Z = 1.974, *p* = 0.048.

## Data Availability

The datasets used and/or analyzed as part of the present study are available from the corresponding author on reasonable request.

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
