# Peer review of "Self-Efficacy Mediates Acculturation and Respite Care Knowledge of Immigrant Caregivers"

_ijerph, 2021, doi:10.3390/ijerph182010595_

Round 1

Reviewer 1 Report

The authors presented interesting work, but the discussion needs improvement.

A more detailed comparison of own results with those of other researchers is recommended. It is proposed to separate from the section discussion, subsection limitations and implications of research results for professional practice. Please specify the conclusions of the research.

Reviewer 2 Report

This study aimed to investigate relationships among acculturation, self-efficacy, and respite care knowledge in immigrant caregivers, and to determine the mediating effects of self-efficacy on the relationship between acculturation and respite care knowledge.

The abstract is too long. It shold be more concise. Avoid keywords like Background, methods etc in the abstract. Please, look at other papers publish in this journal and follow that model of abstract.

A separate section for literatire review is required.

Highlight in the introduction the novelty of your study compared to previous studies.

Is the sample representative to the population?

A section with the mathematical description of the method is necessary.

Pearson’s correlation coefficient is computed for data with normal distribution. You should check this before.

Explain what you have in table 2: F (p)/t (p)/r (p).

The section of conclusions should be extended. Mention limits of the study, future directions of research.

Add more refereces from literature.

Reviewer 3 Report

The results are rare. Tables are shown, but mostly (removing the sociodemographic data is not commented).

If the study was conducted in 2019, how did COVID-19 affect the study and the data collection?

It is necessary to indicate it. Indicate the months of data collection, indicates 2019, but not the period. 

Round 2

Reviewer 2 Report

The paper could be published. 

Reviewer 3 Report

Dear authors, Thanks for reconsidering my reviews, I believe that after the modifications, the manuscript contributes great interest to scientific knowledge,

Sincerely